# Proton and molecular permeation through the basal plane of monolayer graphene oxide

Z. F. Wu[1,2,9], P. Z. Sun[3,9] ✉, O. J. Wahab [4,9], Y. T. Tan[1], D. Barry[1],
D. Periyanagounder[1,2], P. B. Pillai [2,5], Q. Dai [1,2], W. Q. Xiong [6], L. F. Vega [7,8],
K. Lulla[2], S. J. Yuan [6], R. R. Nair [2,5], E. Daviddi [4], P. R. Unwin [4] ✉,
A. K. Geim[1,2] ✉ & M. Lozada-Hidalgo [1,2,8] ✉

Two-dimensional (2D) materials offer a prospect of membranes that combine negligible gas permeability with high proton conductivity and could outperform the existing proton exchange membranes used in various applications including fuel cells. Graphene oxide (GO), a well-known 2D material, facilitates rapid proton transport along its basal plane but proton conductivity across it remains unknown. It is also often presumed that individual GO monolayers contain a large density of nanoscale pinholes that lead to considerable gas leakage across the GO basal plane. Here we show that relatively large, micrometer-scale areas of monolayer GO are impermeable to gases, including helium, while exhibiting proton conductivity through the basal plane which is nearly two orders of magnitude higher than that of graphene. These findings provide insights into the key properties of GO and demonstrate that chemical functionalization of 2D crystals can be utilized to enhance their proton transparency without compromising gas impermeability.

Defect-free graphene is impermeable to all atoms and molecules[1,2] but allows relatively easy permeation of thermal protons[3–5]. The latter finding was unexpected and contradicted theoretical calculations that suggested insurmountable barriers for proton transsport[6,7]. This led to speculation that accidental nanoscale holes were fundamentally necessary for the proton conductivity through graphene[8–10]. Those conclusions were based on experiments using chemical-vapor-deposited graphene that often displays such pinholes[8,10]. The controversy was resolved only recently, with the demonstration that proton permeation through defect-free graphene is spatially nonuniform and occurs predominantly through wrinkles and nanoripples[11]. The morphological distortions of the perfect graphene lattice induce

local strain and curvature, which considerably reduces the energy barrier for protons[7,11,12]. This finding raises the question of whether intentionally induced lattice distortions[11,13] can be employed to enhance the proton conductivity of 2D materials. In this context, graphene oxide, a chemical modification of graphene, offers a convenient testing ground. GO's surface is covered with hydroxide, epoxide and other functional groups that are covalently bonded to the graphene lattice and induce nano- and atomic-scale roughness[14], that is, a high density of morphological distortions.

GO has already attracted considerable interest as a proton conductor because of its high lateral proton conductivity along the basal plane, which is facilitated by surface functional groups[15–18].

[1]Department of Physics and Astronomy, The University of Manchester, Manchester M13 9PL, UK. [2]National Graphene Institute, The University of Manchester, Manchester M13 9PL, UK. [3]Institute of Applied Physics and Materials Engineering, University of Macau, Avenida da Universidade, Taipa, Macau 999078, China. [4]Department of Chemistry, University of Warwick, Coventry CV4 7AL, United Kingdom. [5]Department of Chemical Engineering, The University of Manchester, Manchester M13 9PL, UK. [6]Key Laboratory of Artificial Micro- and Nano-structures of the Ministry of Education and School of Physics and Technology, Wuhan University, Wuhan 430072, China. [7]Research and Innovation Center on CO2 and Hydrogen (RICH Center) and Chemical Engineering Department, Khalifa University, PO Box 127788 Abu Dhabi, United Arab Emirates. [8]Research and Innovation Center for graphene and 2D materials (RIC2D), Khalifa University, PO Box 127788 Abu Dhabi, United Arab Emirates. [9]These authors contributed equally: Z. F. Wu, P. Z. Sun, O. J. Wahab. ✉e-mail: pengzhansun@um.edu.mo; p.r.unwin@warwick.ac.uk; geim@manchester.ac.uk; marcelo.lozadahidalgo@manchester.ac.uk

This in-plane conductivity makes GO a promising additive to Nafion[19–21] and fosters interest in proton-conductive multilayer GO films, so-called GO laminates, which have been extensively examined for their potential as proton-conducting membranes[18,21,22]. However, little is known about proton and molecular transport across the basal plane of GO monolayers (usually obtained by the Hummers' method[23,24]). It is universally assumed that there are many nanoscale holes within individual GO monolayers, which enable considerable proton and gas flows[17,25–28]. In this work, we show that high-quality GO monolayers obtained by the Hummers' method contain large areas that are as impermeable to helium (smallest of all atoms) and other gases as defect-free graphene. The same membranes display proton permeabilities surpassing that of graphene by a factor of ∼50 and approaching those found for monolayers of hexagonal boron nitride (hBN)[3] and mica[29] which are the 2D materials with the highest proton permeability known so far. Using scanning electrochemical cell microscopy, we show that the enhanced proton transport takes place over a large number of active sites that can be attributed to carbon-oxygen bonds that locally distort the underlying graphene lattice of GO.

## Results

### Gas permeation experiments

GO monolayer flakes with large lateral sizes of 10 to 30 μm were synthesized by the Hummers' method and exfoliated using short-duration ultrasonication and step-wise separation, as reported previously[30] (Preparation of GO monolayers in Methods). The oxidation process yielded monolayer crystals that displayed a prominent D band in their Raman spectra (Supplementary Fig. 2). These monolayers were used to prepare membranes for all the experiments described below. In the first set of measurements, GO monolayers were suspended over apertures that were etched through silicon-nitride wafers and had a diameter of ∼2 μm (Supplementary Fig. 3a). The resulting membranes were first characterized by atomic force microscopy (AFM) and only devices with no visible imperfections (e.g., cracks and folds) were studied further (Supplementary Figs. 2a-c). The membranes were then tested using a helium-leak detector that could detect gas flows down to ∼$10^8$ atoms s$^{-1}$, which would be sufficient to examine Knudsen flows through a single pinhole of 1 nm in size ('Helium leak testing of GO monolayers', Supplementary Fig. 3b). No leakage could be found through the studied GO membranes without AFM-visible damage (total examined area of >30 μm$^2$). To the best of our knowledge, He leak tests of GO monolayers were not reported in earlier literature, probably due to challenges associated with making suspended GO membranes that are more fragile than those made from graphene.

The absence of nanoscale pinholes discernable by He-leak detectors makes GO essentially gas impermeable for most purposes. However, these tests could not rule out the presence of smaller, vacancy-like defects in the underlying graphene lattice of GO. Such defects are expected to exhibit thermally activated permeation and, although they are practically impenetrable for large gas molecules, small helium atoms can pass through with rates of ∼$10^3$–$10^4$ atoms s$^{-1}$ under a pressure of 1 bar[31]. This means that, in principle, many angstrom-scale defects could be present in our GO membranes but remained indiscernible using the He-leak detector. To check for such angstrom-scale pinholes, we employed a recently developed technique that can sense gas flows of as little as a few atoms per hour, that is, provides more than 10 orders of magnitude higher sensitivity than that of the best helium-leak detectors[2,31]. The technique is based on micrometer-size wells etched into hBN or graphite monocrystals, which are then sealed with 2D crystals[1,2] (Fig. 1a, Supplementary Fig. 4; Gas permeation measurements using microcontainers in Methods). For our experiments, we fabricated several hundreds of such microwells and sealed them with GO monolayers. The resulting microcontainers were carefully examined optically and by AFM (Fig. 1b). Most (>90%) of the sealing membranes were found broken,

presumably due to strain-induced during fabrication procedures and, especially, at sharp edges of the microwells. Devices with any AFM-visible defects in either GO membranes or their 'atomically tight' sealing[2] were discarded.

The microcontainers that passed the above AFM scrutiny were placed inside a chamber filled with a pressurized gas. If the GO monolayer were permeable, the gas atoms would gradually accumulate inside the microcavity[31,32]. Accordingly, after being taken out of the chamber, the pressure inside the microcontainer would be higher than outside, causing the GO membranes to bulge. We carefully monitored the bulging with AFM and quantified it by tracking the lowest position, $\delta$, in the membrane's AFM profile (Fig. 1b). The data in Fig. 1c show changes in $\delta$ with respect to the initial position $\Delta\delta = \delta(t) - \delta(0)$ as a function of the time, $t$, that the devices spent under 1 bar of helium (kinetic diameter of 2.6 Å). For most of the devices (∼70%), no $\Delta\delta$ could be noticed over a one-month period within our experimental accuracy for $\Delta\delta$ of better than 1 nm. This demonstrates that those GO membranes were completely impermeable to He, similar to pristine (non-oxidized) graphene[2,31]. Note that, if a single angstrom-scale defect were present in our GO membrane, it would rapidly inflate and deflate by tens of nm, as reported previously[31]. The remaining 30% of the microcontainers without AFM-visible pinholes either exhibited some bulging or failed completely. For completeness, we performed similar experiments exposing our microcontainers to other inert gases (Ne, Ar, Kr and Xe) and reached the same conclusions (Fig. 1d).

The finding that GO monolayers can pass our stringent gas permeability tests is unexpected. On one hand, this seemingly contradicts many observations by transmission electron microscopy. These experiments report such a high density of nanoscale pinholes in GO[26,33] that every single one of our devices should be expected to fail the tests, even by the standard He-leak detector. However, those pinholes visible by electron microscopy were likely introduced and/or enlarged by the exposure to high-energy electrons that damage the graphene lattice considerably weakened by functional groups[26]. On the other hand, our results are consistent with the fact that certain chemical procedures can gently remove functional groups from GO, reducing it to practically pristine graphene exhibiting little D peak[34]. The latter evidence suggests that the presence of functional groups covalently attached to graphene does not necessitate the creation of pinholes in the underlying graphene lattice. From our experiments using the He-leak detector, we estimate that regions of GO as large as 30 μm$^2$ in size can be free from pinholes larger than 1 nm in diameter. As for atomic-scale defects, our results show that there exist micrometer-size areas of GO without such defects, but it is difficult to quantify their occurrence probability. Indeed, the low success rate of making completely impermeable GO membranes could be due to either the fragility of the functionalized graphene lattice as a whole or because angstrom-scale defects make the lattice around them more fragile and only membranes without such defects survived the fabrication procedures.

### Proton transport through GO monolayers

Having established that gas-permeable defects were absent in GO membranes without AFM-visible damage, we proceeded to examine proton transport through GO. We used the same devices that passed the impermeability test with the He-leak detector described in the previous section. These membranes were coated on both sides with a proton-conducting polymer (Nafion) and electrically connected with proton-injecting electrodes (Fig. 2a, inset). Details of the fabrication procedures can be found elsewhere[3,35]. For comparison, we made similar devices but, instead of GO, used mechanically exfoliated (pristine) graphene that was not subjected to any oxidation processes.

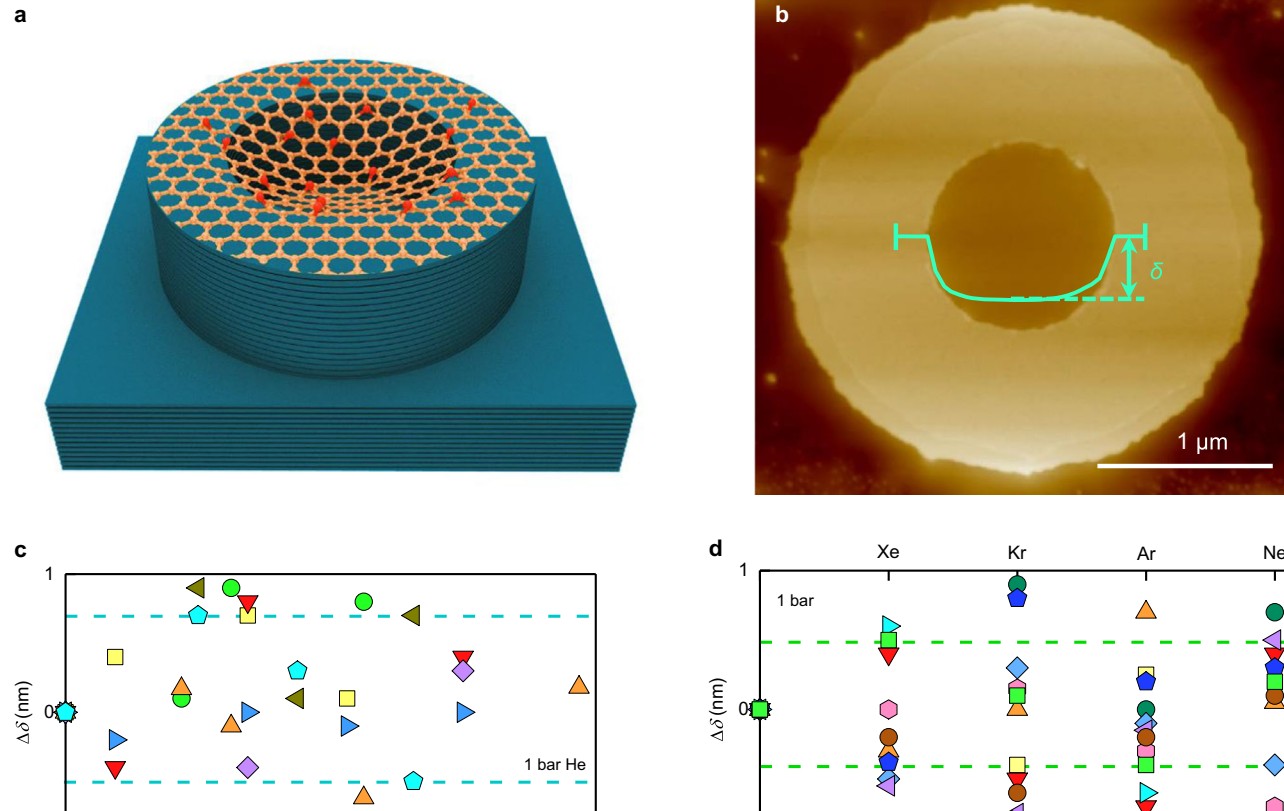

**Fig. 1 | Gas impermeability of GO monolayers. a** Schematic of microcontainer devices. **b** AFM image of a typical device. Green curve, profile along the membrane's diameter. The vertical bars on the green curve mark the 150 nm region around the diameter over which such profiles were averaged. Note that $\delta$ was typically a few tens of nm so that, for clarity, the vertical scale for the green curve is exaggerated with respect to the well's micrometer diameter. The sagging inside is due to van der Waals interaction with the sidewalls[2,31]. **c** Changes in $\delta$ for GO-sealed microcontainers placed under 1 bar of He for a month (different symbols correspond to 11 different devices). **d** Similar measurements for 8 GO-sealed microcontainers placed under 3 bar of Xe, Kr, Ar and Ne. The dashed lines in panels c and d show the full-range scatter for the devices coded with the same colors.

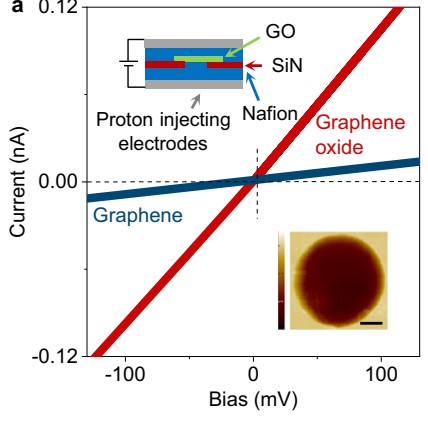

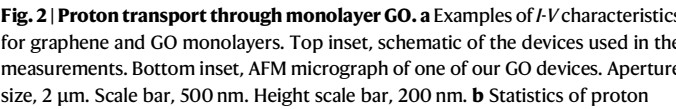

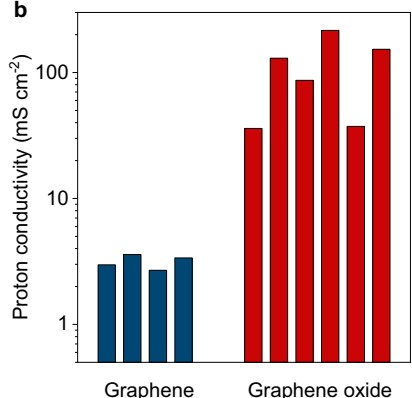

**Fig. 2 | Proton transport through monolayer GO. a** Examples of *I-V* characteristics for graphene and GO monolayers. Top inset, schematic of the devices used in the measurements. Bottom inset, AFM micrograph of one of our GO devices. Aperture size, 2 μm. Scale bar, 500 nm. Height scale bar, 200 nm. **b** Statistics of proton conductivity for graphene and GO monolayers. Each bar represents a different device. Our GO devices exhibited greater variability (factor of ~5) compared to pristine-graphene ones (factor of ~2). The difference is attributed to the uneven distribution of functional groups on the GO surface[25,26].

For electrical measurements, the devices were placed inside a chamber filled with humid hydrogen to ensure the high proton conductivity of Nafion (Proton conductivity measurements in Methods).

For both pristine graphene and GO, we found the measured currents to vary linearly with applied bias (Fig. 2a), and the extracted proton conductivities for different devices are shown in Fig. 2b. For

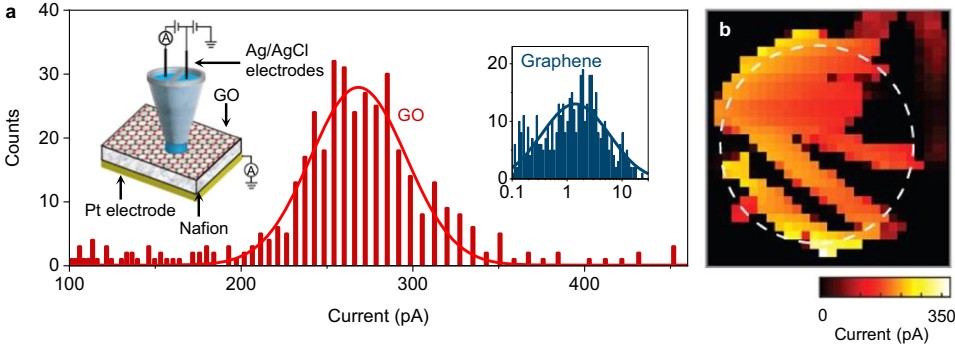

**Fig. 3 | Distribution of proton currents through GO membranes. a** Statistical distribution of the proton current values measured over the entire scanned area in panel b. Right inset, similar statistics for a similar device where, instead of GO, pristine graphene was used[11]. Left inset, schematic of our SECCM setup. For details, see Supplementary Fig. 5 and Methods. **b** Example of SECCM maps for monolayer GO. The white circle marks the 2 µm diameter aperture in silicon-nitride. Each pixel is 100 × 100 nm in size.

pristine graphene, the room-temperature conductivity was found to be ~2-3 mS cm$^{-2}$, in agreement with the earlier reports[3,5,11]. In contrast, GO monolayers were notably more transparent to protons, exhibiting conductivity of $110 \pm 70$ mS cm$^{-2}$, which is on average ~50 times higher than that of pristine graphene (Fig. 2b) and comparable to that of monolayer hBN[3] and few-layer mica[29]. Although nanoscale pinholes were ruled out by our He-leak tests, one can still argue that the relatively large proton permeability is not an intrinsic characteristic of GO but occurs through a small number of atomic-scale pinholes, as protons (unhydrated H$^+$) are much smaller than He atoms. To investigate the latter possibility, we examined GO membranes using scanning electrochemical cell microscopy (SECCM), which provided maps of GO's proton conductivity with nanoscale resolution.

### Spatial distribution of proton currents

For SECCM, we followed the methodology described in our recent study of proton transport through graphene and hBN crystals[11]. The devices were similar to those used in the electrical measurements reported in Fig. 2, except that only one (bottom) side of the suspended GO membranes was coated with Nafion and contacted electrically. The top side of the membrane was left uncoated to allow examination with a SECCM probe. The probe consisted of a nanopipette filled with 0.1 M HCl and containing Ag/AgCl electrodes (inset of Fig. 3a, Supplementary Fig. 5). During the measurements, the probe was positioned over the GO membrane, with its meniscus contacting the GO surface, and protons were electrochemically pumped from the nanopipette through GO and collected by the Pt electrode (SECCM setup and scanning protocol in Methods). As the probe's position was changed, a map of the spatial distribution of proton currents through GO was acquired. Figure 3b shows an example of such maps obtained for GO monolayer devices. For areas away from the aperture in the silicon-nitride wafer, only small parasitic currents of ~10 fA could be detected as the silicon-nitride layer blocked protons. In stark contrast, for the areas in which GO was in contact with Nafion, currents were several orders of magnitude higher (Fig. 3, Supplementary Fig. 5b).

The key finding from these measurements is that the maps did not display isolated sites with high proton currents that could be attributed to pinholes, unlike the case of previous experiments using chemical-vapor-deposited (but nonoxidized) graphene that clearly displayed such pinholes[10]. Instead, we observed many sites within the current range of 200-300 pA, which is approximately two orders of magnitude higher than in our reference devices using pristine exfoliated graphene, in agreement with ref. 11 (inset of Fig. 3a). Besides the high activity regions, our SECCM maps also revealed extended regions exhibiting much smaller proton currents within the apertures (dark regions in Fig. 3b). To understand this spatial inhomogeneity, we note that, unlike our work on pristine graphene which found a correlation

between active sites and topography of graphene membranes[11], no such correlations were observed for GO. Accordingly, the low proton currents are attributed to poorly oxidized areas, which is a common feature of GO[25,26]. The microscopic picture provided by SECCM was also validated by integrating the currents over the whole membrane area. This yielded an estimate for GO's average proton conductivity as ~500 mS cm$^{-2}$, which was approximately 5 times larger than the global conductivity reported in Fig. 2b. This disagreement is slightly larger, compared to that between SECCM and global conductivity measurements[11] in the case of pristine graphene, which differed by a factor of ~3. We attribute the divergence to the fact that GO's high hydrophilicity resulted in a smaller contact angle with respect to non-oxidized (pristine) graphene[36], which led to a bigger meniscus on the GO surface (exceeding the SECCM tip diameter) and hence an overestimation of the areal conductance (Supplementary Fig. 6, SECCM setup and scanning in Methods section).

## Discussion

Our experiments show that for areas without pinholes, the basal plane of GO is completely impermeable to all atoms and molecules, similar to pristine graphene. At the same time, GO exhibits notably higher permeability to protons than graphene. The large number of active sites that show large proton currents rules out isolated pinholes as the reason for the enhanced proton permeability. We attribute the enhancement to microscopic corrugations of the underlying graphene lattice, which are caused by functional groups bonded to the GO surface[14]. Indeed, it has recently been shown that proton permeation through pristine graphene monolayers is strongly facilitated by nonflat regions such as nanoscale ripples where local strain reduces the energy barrier for proton permeation[2,7,11]. The functional groups on GO's surface also create microscopic distortions, acting somewhat like adsorbates that are known to cause nanoripples on pristine graphene[2,37–39]. Our density functional theory calculations (Supplementary Fig. 7) show that such nanoripples can indeed lower the barrier for proton transport by ~20−50%, similar to nanoscale corrugations in graphene[11]. Further theoretical analysis to assess the role of such distortions would be beneficial but is impeded by the lack of microscopic details about GO functionalization, which remain under debate and vary for different production recipes[14,25,26,40,41].

The finding that graphene oxide contains micrometer-size areas where the underlying graphene lattice remains intact, despite strong distortion by functional groups, is critically important for analyzing future experiments involving GO. While high in-plane proton conductivity of GO was already known, our work has revealed its considerable out-of-plane proton transparency combined with He-gas impermeability. These properties suggest a possible use of GO as, for example, an additive to proton-conducting polymers, enabling thinner

yet more conductive proton exchange membranes, while retaining gas impermeability similar to that of the standard membranes. The reported direct characterisation of individual GO crystals, as opposed to inferring their permeability properties from collective performance in GO powders and laminates, represents a necessary step toward designing and implementing such membranes. Our results also suggest that the proton permeability of other 2D materials can be enhanced using functionalization, which could expand their potential applications in hydrogen-related technologies.

## Methods

### Preparation of GO monolayers

Graphite oxide was obtained using the Hummers' method and the material was exfoliated into monolayers by short-duration sonication in water (3 minutes under 40 W power). Large monolayers (>10–20 μm) were separated from smaller ones using centrifugation procedures[30]. The chemical composition of GO layers was investigated using X-ray photoelectron spectroscopy (XPS) with an ESCA 2SR high-throughput X-ray photoelectron spectrometer (*Scienta Omicron GmbH*). The instrument is equipped with a monochromated Al Kα source (1486.6 eV, 20 mA emission at 300 W), an Argus CU multi-purpose hemispherical analyser and an electron flood gun for less conductive samples. All the peaks were calibrated using the C1s photoelectron peak at ~284.8 eV for graphitic carbon. Spectral deconvolution was performed using CASAXPS (www.casaxps.com) with Shirley-type backgrounds. Supplementary Fig. 1 shows the high-resolution XPS spectra of the C1s region for a lamellar GO film, which indicates a considerable degree of oxidation with five components that correspond to carbon atoms bonded to different oxygen functional groups. These are[30,42–44]: non-oxygenated ring $C = C$ (~284.8 eV), hydroxyl group C-OH (~285.6 eV), epoxy group O-C-O (~286.9 eV), carbonyl $C = O$ (~287.8) and carboxylate $O-C = O$ (~288.8 eV). The inset of Supplementary Fig. 1 shows the survey spectrum for the same sample. The spectrum is dominated by carbon and oxygen. Using high-resolution scans of C1s and O1s regions, we have evaluated the C/O ratio as ~3.5.

To isolate GO monolayers, a droplet of a dilute GO solution containing the resulting large flakes (concentration of ~0.1 mg L$^{-1}$) was drop-cast onto an oxidized silicon wafer covered with a bilayer polymer film (polypropylene carbonate/polyvinyl alcohol) that was used as a sacrificial layer when transferring selected flakes to make the final devices. The deposited GO flakes were examined using both optical and atomic force (AFM) microscopes to check their thickness and quality. Supplementary Fig. 2 shows that our typical GO flakes had a lateral size of a few tens of micrometers and were about 1 nm thick, in agreement with the previous reports for GO monolayers' AFM thickness[45]. Areas within the flakes without visible defects such as wrinkles[46], tears or cracks were chosen for device fabrication (see Supplementary Fig. 2). When handling GO, we avoided heat as it could reduce GO at temperatures >150 °C[26]. We also avoided ultraviolet exposure, which could initiate chemical reactions and create defects in GO's crystal lattice[26].

### Helium leak testing of GO monolayers

For these measurements, circular apertures (2 μm in diameter) were etched into silicon-nitride/silicon substrates (500 nm of silicon-nitride) using photolithography, wet etching and reactive ion etching, as reported previously[3] (Supplementary Fig. 3a). GO monolayers were deposited over the apertures using the dry transfer method[47]. The resulting membranes were examined by AFM and those with any visible damage were discarded. To rule out nanoscale defects invisible to AFM, we tested the membranes with respect to their helium permeability using leak detector *Leybold Phoenix L300i*. To this end, suspended membranes were exposed to a He pressure of up to 1 bar on one side while the other side faced a vacuum chamber equipped with the leak detector. We found that GO monolayers were impermeable within the instrument's accuracy of ~10$^8$ He atoms s$^{-1}$ (red curves in Supplementary Fig. 3b). To appreciate this level of sensitivity, the figure also shows helium-flow rates for a single pinhole of 50 nm in diameter. The flow rates reach ~10$^{13}$ atoms s$^{-1}$ at 1 bar, in quantitative agreement with the Knudsen theory[2,31]. This allows us to estimate that, for a single pinhole of 1 nm in diameter, gas flows should be above 10$^9$ helium atoms per second if the leak rates scale proportionally to the aperture's area as they should in the Knudsen regime. This flow rate is an order magnitude higher than our experimental resolution and, accordingly, the helium leak detector should have enough sensitivity to detect even a single 1-nm hole if it were present in our GO membranes.

### Gas permeation measurements using microcontainers

The fabrication procedures for making monocrystalline micro-containers are described in detail in refs. 2,31. In brief, polymer rings with inner and outer diameters of 0.5–1 μm and 1.5-2 μm, respectively, were defined by e-beam lithography on top of atomically flat areas (free of terraces) of cleaved graphite or hBN monocrystals. The rings were used as a mask to dry-etch the crystals and form an array of microwells that were ~80 nm deep (Supplementary Fig. 4). The resulting structures were annealed at 400 °C in a H$_2$/Ar atmosphere overnight to remove any possible polymer residues. Then the micro-wells were covered by GO crystals to form tightly sealed micro-containers as illustrated in Fig. 1 of the main text. Such microcontainers were placed inside a gas chamber containing pressurized inert gases (Xe, Kr, Ar, Ne and He) for typically several days. After this, the microcontainers were taken out of the chamber into the ambient atmosphere and rapidly (within minutes) examined using AFM in the PeakForce mode to detect any changes in profiles of the suspended GO membranes.

### Proton conductivity measurements

To measure proton conductivity of GO, both sides of the freestanding membranes were coated with a Nafion solution (5% Nafion; 1100 EW). Proton injecting electrodes were carbon cloth with Pt-on-carbon acting as a catalyst. The devices were measured in a humid hydrogen atmosphere (10% H$_2$ in Ar, 100% humidity), and the *I-V* curves were recorded using Keithley SourceMeter 2636 A. Measurements were limited to $T < 60$ °C to prevent GO membranes' rupture. We emphasize that, despite the relatively high conductivity of GO membranes, our devices' resistance was still ~100 times higher than that of devices with empty apertures (for example, if a membrane was removed)[3]. This assured that the series resistance arising from a finite conductivity of Nafion could be neglected and we measured the intrinsic proton properties of GO monolayers.

### SECCM setup and scanning protocol

Devices for SECCM were similar to the suspended GO membranes used for the proton conductivity measurements, except their top side was left accessible to the SECCM probe. The bottom side coated with Nafion was electrically connected to a Pt electrode[48].

The SECCM probe consisted of a quartz theta pipette with a tip opening diameter of ~200 nm as previously reported[11]. The nanopipette was filled with 0.1 M HCl electrolyte and two Ag/AgCl quasi-reference counter electrodes were used to electrically connect each of the pipette channels[49]. The SECCM was performed using a home-built workstation[50] enclosed in a Faraday cage with heat sinks and vacuum panels to minimize noise and thermal drift. Two home-built electrometers were used for current measurements, together with 8$^{th}$-order brick-wall filters with the time constant for current amplifiers set at 10 ms.

SECCM measurements were performed in the hopping mode[48,51,52]. In this case, the probe approaches the sample until the

meniscus at the end of the tip makes contact with the surface. Upon com-completion of a measurement on one site, the probe is retracted and 'hops' to the next site. Two voltage controls are used, as explained extensively in ref. 11. In brief, the first is the potential $E_{bias}$ between the two quasi-reference counter electrodes, which drives an ionic current between the channels in the nanopipette (Supplementary Fig. 5a). This potential serves as a feedback signal to detect contact between the meniscus and sample surface. The second voltage $E_{app}$ between the collector electrode and the SECCM probe drives the electrochemical proton reduction at the Pt electrode, yielding the current $I_{collector}$. During measurements at each site, $I_{collector}$-$t$ curves are acquired. They display resistor-capacitor decay characteristics and achieve a steady state typically within ~400 ms after contact with the sample surface. The data presented in the maps is the average over the last 100 ms for the measured $I_{collector}$-$t$ curves (Supplementary Fig. 5b). $E_{app}$ is chosen as $E_{collector} = -0.5\,V$ vs Ag/AgCl electrodes, which is equivalent to an overpotential of ~0.2 V vs the standard potential for the hydrogen evolution reaction.

While the focus of the SECCM studies is the measurement of proton permeability via $I_{collector}$, the current that flows between the 2 channels of the theta nanopipette ($I_{DC}$) due to $E_{bias}$ provides qualitative information on the local degree of wetting[11]. Upon the approach of the nanopipette to the surface, there is a characteristic 'jump-to-contact' as the meniscus wets the surface, resulting in an abrupt change in $I_{DC}$ (at time = 0.5 s in the inset Supplementary Fig. 6a). The magnitude of this current change, compared to the ('float') current when the probe is retracted, is much larger for GO than for graphene (Supplementary Fig. 6a), indicating a larger meniscus and stronger wetting of the GO surface[53,54] (data are for graphene and GO on the SiN$_x$ support). On the other hand, the broad distribution of 'jump-to-contact' currents for GO is consistent with an inhomogeneous functionalization in the membrane. The wetting behavior discussed so far is mirrored when the probe is withdrawn. In this case (at = 1.0 s in the inset Supplementary Fig. 6a), there is a stretching of the meniscus, resulting in a further increase in current (visible as a spike in Supplementary Fig. 6a), before meniscus contact is broken. The magnitude of this change is much larger and more heterogeneous for GO than graphene (Supplementary Fig. 6b), again consistent with more extensive wetting of GO on the nanoscale.

**Density functional theory calculations**

DFT calculations were performed using the VASP package[55]. The exchange-correlation potential and ion-electron interactions were described by the generalized gradient approximation (GGA) and projected augmented wave (PAW) method[56,57]. The kinetic energy cut-off and the $k$-point meshes were set to 500 eV and 7×7×1, respectively[58]. The van der Waals interactions were considered and treated by the semi-empirical DFT-D2 method[59,60]. All atoms were allowed to fully relax to the ground state and the spin-polarization was included. We used climbing-image nudged elastic band (CI-NEB) methods to search for transition states (TS)[61,62]. The latter correspond to states with the highest total energy $E_{max}$ along the reaction path. Accordingly, the energy barrier $E_b$ was evaluated as the difference between $E_{max}$ and the initial energy $E_{in}$.

To model GO, oxygen functional groups were attached to the graphene lattice. For simplicity, only epoxy groups, that are most abundant in our GO films (Supplementary Fig. 1), were modelled. We considered several epoxy-graphene structures, each containing a different number ($n$) of functional groups. Supplementary Fig. 7a illustrates that these structures are buckled due to $sp^3$-$sp^3$ C-C hybridization. The largest buckling happens if all the carbon atoms within a single hexagon ring are bonded with oxygen ($n = 3$). We then calculated the energy for the proton-GO system as the proton transfers through the distorted hexagon ring. As shown in Supplementary Figs. 7b and 7c, the energy barrier $E_b$ for proton translocation is notably lowered with increasing the number of attached oxygen atoms. At $n = 3$, where all the carbon atoms are saturated by oxygen (Supplementary Fig. 7a), the calculated $E_b$ reaches minimum, which is about half that of the pristine graphene ($n = 0$). The observed trend in $E_b$ is inversely correlated with that in the calculated length of C-C bonds in the buckled graphene lattice (Supplementary Fig. 7c), demonstrating that the pore size changes associated with the structural distortions lower the barrier for proton transport.

## Data availability

All data supporting the key findings of this study are available within the article, the Supplementary Information file and is available at: https://zenodo.org/records/10042004.

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

## Acknowledgements

This work was supported by EPSRC (EP/V047981, P.R.U. & E.D.; EP/V007688/1, P.R.U. & O.J.W.), UKRI (EP/X017745, M.L.-H), The Royal Society (Wolfson Research Merit Award, P.R.U., URF\R1\201515, M.L.-H. & URF\R\191009, R.R.N.), Lloyd's Register Foundation (Nano Grant G0084, A.K.G.), European Research Council (786532-VANDER, A.K.G & 679689, R.R.N.), the Directed Research Projects Program of the

Research and Innovation Center for Graphene and 2D Materials (RIC2D) at Khalifa University (L.V., A.K.G., M.L.-H. and R.R.N.), The Graphene Flagship (Core3 881603 R.R.N.), The Royal Academy of Engineering and Carlsberg Research Laboratory (RCSRF2223-1671, R.R.N.) and the Leverhulme Trust (PLP-2018-220, R.R.N.). XPS was performed at the Henry Royce Institute for Advanced Materials, funded through EPSRC grants EP/R00661X, EP/P025021 and EP/P025498. We thank Dr Ben Spencer for helping with XPS measurements. P.Z.S. acknowledges the FDCT project No. 006/2022/ALC of the Macao Centre for Research and Development in Advanced Materials.

## Author contributions

M.L.-H., A.K.G., P.R.U. and P.Z.S. designed the project and analyzed its results. Z.F.W., D.P. and P.Z.S. fabricated the GO devices. P.Z.S. and Y.-T.T. performed gas permeation measurements. D.B. and Q.D. performed proton transport measurements. O.W. and E.D. performed SECCM measurements. P.B.P. and R.R.N. synthesized GO material. W.Q.X. and S.J.Y. performed DFT calculations. K.L. and L.V. provided technical and theory support, respectively. M.L.-H. and A.K.G. wrote the manuscript with input from all the authors.

## Competing interests

The authors declare no competing interests.
