## [Peer Review File · Nature Communications]

REVIEWERS' COMMENTS

Reviewer #1:

Remarks to the Author:

This manuscript reports on the notably higher proton conductivity through the basal plane of GO for areas without pinholes, which is completely impermeable to all gases. The highlight of this paper is the discovery of relatively large single-layer graphene oxide regions with high proton conductivity and low gas permeability, which provides new ideas for the development of two-dimensional materials for applications such as fuel cells. This work is interesting. However, more additional evidences should be provided to strongly support the statements.

The comments are as follows.

1. The article does not characterize and analyze the structure and functional groups of graphene oxide monolayers in detail, such as XPS, Fourier transform infrared spectroscopy, X-ray diffraction, etc., which can better reveal the oxidation degree, functional group type and distribution, lattice distortion and other information, and establish a clearer connection with proton conductivity.
2. The article does not systematically investigate the proton conductivity of graphene oxide monolayers under different temperature, humidity, pH value and other conditions, which may affect the stability of functional groups, proton concentration and mobility, etc., thus affecting proton conductivity. In addition, the article does not compare with other two-dimensional materials or composite materials to show the advantages and potential of graphene oxide monolayers in the field of proton exchange membranes.
3. The article does not perform theoretical calculations and simulations on the proton conductivity of GO monolayers, such as density functional theory, molecular dynamics, etc., which can explain and predict the influence of functional groups on the proton penetration barrier from the atomic scale, as well as the proton transport path and rate on the surface and inside of GO.
4. The authors should further provide data on proton transport, such as the effect of proton concentration.
5. The authors attribute the differences in the data collected in Figure 2b to different oxidation levels for different GO areas. How does the author exclude the accidental factors of experimental error?
6. It is worth noting that the GO exhibit notably higher permeability to protons than graphene, while the reason for this high permeability performance is not well explained.
7. The authors claimed the monolayer graphene oxide has been prepared. The performance should be compared with monolayer graphene or graphite.
8. Long-term stability is one of the important parameters for membrane performance.
9. References form should be check. For example, authors, .and Journal Abbreviations. For example, Ref. 10, 20, 24 and 25.
10. Some important and newest work should be cited. For example, Nature Nanotech 2017,12,509; Nature, 2021, 596, 2021; Nat. Commun.2023,14, 2580; et al.

Reviewer #2:

Remarks to the Author:

Recommendation: Minor revisions

In this manuscript, Z. F Wu and co-authors demonstrated the proton and molecular permeation through the basal plane of monolayer graphene oxide. The work is interesting and I would

recommend its publication. However, there are still some questions that need to be demonstrated clearly. Given this, before this paper is accepted for publication, the author should address the following issues.

i) As Nafion is coated on both sides of the GO in the proton conduction measurement, is there any effect/interference of Nafion's proton conductivity in the resultant proton conduction of graphene or GO monolayer? How about the proton conductivity of a bare Nafion-prepared conductivity measurement cell (without the use of graphene or graphene oxide)?

ii) In the manuscript, there is no information about the temperature dependence of the proton conductivity of the GO monolayer. The activation energy for proton conductivity can be obtained from the Arrhenius plot of the temperature dependence of proton conductivity, which might provide significant information regarding the mechanism/conduction pathway of 'through the basal plane of the monolayer of graphene oxide'.

iii) It seems an error in the scale bar of Supplementary Figures 1 c and e. As the author describes in the figure caption that the scale bar in Figure 1c is 20 μ m, then the horizontal scale bar in Figure 1e should not be 0.6 μ m.

Reviewer #3:

Remarks to the Author:

This manuscript reported a study on the proton and gas molecular permeation behavior through the basal plane of monolayer graphene oxide. The authors prepared micrometer-scale monolayer GO and found that the proton transport takes place over the active sites of carbon-oxygen bonds that distort the graphene lattice of GO, while no gas can transport through the GO sheet. The above findings are interesting and helpful to understand the transport mechanism of proton through pinhole-free GO sheets. The authors carefully conducted precise experiments to test the gas leakage which gives more convincing proof and provides a good example for precise measurement.

The work is of good scientific contribution to the development of GO materials, one thing I'm concerned about is that when graphene oxide is used in hydrogen fuel cell, is there a risk that it will be possibly reduced to some extent and subsequently results in some operational safety problem, especially for a long-term run? Could the authors give a prospective comment on the real application possibility of GO membrane in proton-conducting fuel cell.

Reply to comments of Reviewer #1

This manuscript reports on the notably higher proton conductivity through the basal plane of GO for areas without pinholes, which is completely impermeable to all gases. The highlight of this paper is the discovery of relatively large single-layer graphene oxide regions with high proton conductivity and low gas permeability, which provides new ideas for the development of two-dimensional materials for applications such as fuel cells. This work is interesting. However, more additional evidences should be provided to strongly support the statements.

We thank the Reviewer for this positive assessment of our work and taking his/her time to evaluate the paper in such detail. We also appreciate the Reviewer's thoughtful analysis and suggestions about how to improve the work. We did our best to address all the Reviewer's comments as discussed below.

The comments are as follows.

1. The article does not characterize and analyze the structure and functional groups of graphene oxide monolayers in detail, such as XPS, Fourier transform infrared spectroscopy, X-ray diffraction, etc., which can better reveal the oxidation degree, functional group type and distribution, lattice distortion and other information, and establish a clearer connection with proton conductivity.

The GO used in this work was obtained using the fabrication procedures routinely employed in one of the collaboration groups. The resulting material was previously characterized by many techniques, including by FTIR and XPS analysis (e.g., Nature 559, 236–240 (2018), Nat. Materials 16, 1198–1202 (2017)). We are sorry that we overlooked mentioning this characterization in the original manuscript. The Reviewer is right that this can be important for readers. Following the Reviewer's advice, we have included the corresponding references in the revised manuscript and, for completeness, also provided XPS data for our particular batch of GO (see new Fig. S1).

2. The article does not systematically investigate the proton conductivity of graphene oxide monolayers under different temperature, humidity, pH value and other conditions, which may affect the stability of functional groups, proton concentration and mobility, etc., thus affecting proton conductivity. In addition, the article does not compare with other two-dimensional materials or composite materials to show the advantages and potential of graphene oxide monolayers in the field of proton exchange membranes.

The primary aim of this report has been to show that GO monolayers contain areas permeable to protons but completely impermeable to gases. This property was unexpected from the previous literature, and proving this point required huge effort involving two years of work and three different techniques: nano-balloons, proton transport measurements and scanning electrochemical cell microscopy. Taken separately, none of them was sufficient.

We agree with the Reviewer that a systematic investigation of GO's proton permeability at different pH, humidity or temperature would be useful. However, one cannot possibly cover all the ground in just one paper, whereas tens of thousands of papers and counting have been published about GO. We believe that the current results are of interest for the large community working on GO and should be made available soon to enable feedback from the community. Moreover, some of the suggested studies would require completely different experimental approaches unavailable at the authors' labs (please see our reply to point 4 below, which concerns pH measurements, and our response to Reviewer 3 about measuring the temperature dependence). We hope that the Reviewer would appreciate that it is not a lack of will but the big difficulties that stop us from carrying such measurements.

As for comparison of different 2D materials in terms of their proton conductivity, following the Reviewer's comment, we have revised the manuscript and provided the suggested comparison. The

data of Figs. 2 and 3 are now discussed more extensively and compared with other proton-conducting 2D crystals such as hBN and mica ('Proton transport through GO monolayers' in the revised main text).

3. The article does not perform theoretical calculations and simulations on the proton conductivity of GO monolayers, such as density functional theory, molecular dynamics, etc., which can explain and predict the influence of functional groups on the proton penetration barrier from the atomic scale, as well as the proton transport path and rate on the surface and inside of GO.

We appreciate this comment. Indeed, such calculations can be informative. In our original manuscript, we had chosen to avoid them because the microscopic details of GO functionalization remain under debate and vary for different production methods (e.g., refs. 14,25,26,40,41). The Reviewer's comment persuaded us to change our opinion, and we now report DFT analysis for the role played by functional groups on GO. Our calculations have revealed that oxygen atoms bonded to graphene strongly distort its crystal lattice. This leads to an expansion of the carbon-carbon bond length, which in turn results in a notable reduction of the energy barrier for proton permeation, of up to 50%. These calculations are now included in Fig. S7 and 'Density functional theory calculations' in Methods.

4. The authors should further provide data on proton transport, such as the effect of proton concentration.

We agree that characterising the effect of proton concentration would be useful. The problem with such a study is that it would require measuring the devices using liquid electrolytes, instead of Nafion. In the reported experiments, Nafion provided not only the source of protons but also mechanical support and stability. Measurements of suspended monolayers inside electrolytes are technically challenging. We have previously done this – with great difficulties – using pristine graphene and monolayer hBN, but suspended GO monolayers have much lower mechanical strength. Following the Reviewer's comment, we again attempted such measurements for GO but our devices were destroyed too quickly to perform a reliable data set. Unfortunately, we cannot see any feasible way to provide the suggested additional measurements in the current setup. Their absence however does not overturn our main result of demonstrating that proton transport occurs through pinhole-free areas in GO. We hope for the Reviewer's understanding about this technical difficulty.

5. The authors attribute the differences in the data collected in Figure 2b to different oxidation levels for different GO areas. How does the author exclude the accidental factors of experimental error?

The experimental accuracy of our measurements using micrometre-sized 2D crystal devices coated with Nafion has been established in earlier reports. Basically, our measurements are accurate within a factor of ~ 2 , as shown using statistics for many identical devices made from various 2D crystals such as graphene or hBN (refs. 3,29). Previously, we have never encountered the variability by a factor of ~ 5 as in the current paper using GO. Therefore, it seems sensible to attribute this scatter to heterogeneity of the oxidation of the crystal in the microscale, which is a known effect in GO. Following the Reviewer comment, we have explained this point in the caption of Fig. 2 in the main text.

6. It is worth noting that the GO exhibit notably higher permeability to protons than graphene, while the reason for this high permeability performance is not well explained.

We appreciate this comment, and the revised manuscript tries to better explain the reason behind the permeability. Basically, proton transport through pristine (non-oxidised) graphene occurs predominantly through wrinkles, nanoripples and other distortions of graphene's crystal lattice, which induce local strain and curvature (ref. 11). This reduces the energy barrier for protons' permeation, a conclusion supported by both experiment and DFT calculations. Since the functional groups (e.g.,

hydroxide) in GO are known to induce nano- and atomic- scale roughness (e.g., ref. 14), we attributed the enhanced permeability to such morphological distortions. This explanation is supported by our latest DFT analysis for proton transport through GO, which is now included in the revised manuscript as mentioned in point 3 above.

7. The authors claimed the monolayer graphene oxide has been prepared. The performance should be compared with monolayer graphene or graphite.

GO's performance was compared to that of (non-oxidised) graphene in Fig. 2 and 3 in the original manuscript. Following the Reviewer comment, this comparison is now discussed more extensively and supplemented by comparison with hBN and mica in 'Proton transport through GO monolayers' in the revised main text.

8. Long-term stability is one of the important parameters for membrane performance.

We agree. To characterise the proton conductivity of our suspended monolayers, we typically measured the devices for several hours continuously, following the established protocol for this type of devices (ref. 3). Longer term stability of 2D membranes is difficult to assess, particularly in the case of GO. Our devices were usually found broken after a couple of days, which we attribute to the lower mechanical stability compared to graphene, as discussed above. Strain induced by variations in temperature or humidity and by creep of Nafion is likely to be responsible for the observed rupture of our suspended GO monolayers. Such ruptures do not seem to happen for non-suspended GO placed either on a surface or inside Nafion (refs. 18,21,22). Given the limitations of our setup using suspended membranes, it would be inappropriate to speculate about long-term stability for other situations, especially setups aimed at applications where suspended membranes are not used. Accordingly, we have focused on the question of whether a single layer of GO is permeable to protons in the absence of pinholes, rather than on the demonstration of potential GO-based applications. Following the Reviewer's comment, we have added this info about stability to the revised manuscript.

9. References form should be check. For example, authors, .and Journal Abbreviations. For example, Ref. 10, 20, 24 and 25.

Very much appreciated. Corrected.

10. Some important and newest work should be cited. For example, Nature Nanotech 2017,12,509; Nature, 2021, 596, 2021;Nat. Commun.2023,14, 2580; et al.

We are sorry for overlooking these interesting and important papers, and the Reviewer's rebuke is much appreciated. These are now cited in the revised manuscript. We hope that we found the right paper: Nature 596, 519 (2021) as the typo above made it ambiguous.

Reply to comments of Reviewer #2

Recommendation: Minor revisions

In this manuscript, Z. F Wu and co-authors demonstrated the proton and molecular permeation through the basal plane of monolayer graphene oxide. The work is interesting and I would recommend its publication. However, there are still some questions that need to be demonstrated clearly. Given this, before this paper is accepted for publication, the author should address the following issues.

We thank the Reviewer for supporting the publication. All the issues raised by the Reviewer are carefully addressed in the revised manuscript.

i) As Nafion is coated on both sides of the GO in the proton conduction measurement, is there any effect/interference of Nafion's proton conductivity in the resultant proton conduction of graphene or GO monolayer? How about the proton conductivity of a bare Nafion-prepared conductivity measurement cell (without the use of graphene or graphene oxide)?

The measured conductivity of 'bare Nafion' devices (same 2 μm diameter devices but without GO) is about 1,300 mS cm^{-2} . This value is ~ 100 times larger than that of our devices with GO, which shows that the reported measurements captured GO's intrinsic proton conductivity and were not limited by Nafion's conductivity. Following the Reviewer's comment, this important point is now explained in 'Proton conductivity' in Methods.

ii) In the manuscript, there is no information about the temperature dependence of the proton conductivity of the GO monolayer. The activation energy for proton conductivity can be obtained from the Arrhenius plot of the temperature dependence of proton conductivity, which might provide significant information regarding the mechanism/conduction pathway of 'through the basal plane of the monolayer of graphene oxide'.

We appreciate this suggestion and agree that temperature dependence measurements would be very informative. In fact, we used such measurements in our previous studies of proton conductivity of pristine graphene (e.g. ref. 3). In the latter case, the measurements were limited to a temperature range from 10 to 60 $^{\circ}\text{C}$ because the induced strain led to rupture of graphene membranes, despite the high strength of graphene. Unfortunately, GO monolayers have much lower mechanical stability (too fragile) and it turned out impossible to measure GO's conductivity over any reasonable interval of temperatures to make Arrhenius plots. We hope that the Reviewer would appreciate why neither we (nor anybody else) know how to resolve this problem for suspended GO monolayers.

iii) It seems an error in the scale bar of Supplementary Figures 1 c and e. As the author describes in the figure caption that the scale bar in Figure 1c is 20 μm , then the horizontal scale bar in Figure 1e should not be 0.6 μm .

We are grateful for spotting this error. It has been corrected in the revised manuscript.

Reply to comments of Reviewer #3

This manuscript reported a study on the proton and gas molecular permeation behavior through the basal plane of monolayer graphene oxide. The authors prepared micrometer-scale monolayer GO and found that the proton transport takes place over the active sites of carbon-oxygen bonds that distort the graphene lattice of GO, while no gas can transport through the GO sheet. The above findings are interesting and helpful to understand the transport mechanism of proton through pinhole-free GO sheets. The authors carefully conducted precise experiments to test the gas leakage which gives more convincing proof and provides a good example for precise measurement.

We appreciate very much this kind assessment of our work and its support.

The work is of good scientific contribution to the development of GO materials, one thing I'm concerned about is that when graphene oxide is used in hydrogen fuel cell, is there a risk that it will be possibly reduced to some extent and subsequently results in some operational safety problem, especially for a long-term run? Could the authors give a prospective comment on the real application possibility of GO membrane in proton-conducting fuel cell.

The thermal reduction of GO in air typically occurs at temperatures exceeding $>150^{\circ}\text{C}$ (ref. 26). While the presence of protons could in principle accelerate this process, several references in the literature show that GO membranes is stable in fuel cells under typical operation conditions. For example, in ref. 16, GO laminates were used as a proton exchange membrane that allowed stable operation for 100 hrs. In terms of safety, the reduction of GO in the presence of protons is expected to form water, rather than H_2 gas, which probably would not represent a safety concern.

As for GO's prospects in real applications, we believe that the material does have a potential to improve the existing proton membranes. Lateral (in-plane) proton conductivity of GO is very high and, as shown in our report, proton transport in the out-plane direction is also considerable. Combined with GO's impermeability to gases, the material could be added to, e.g., Nafion. This may enable, for example, thinner proton exchange membranes that would be more conductive than standard ones, while preserving their impermeability to gases. Understanding GO's detailed properties is an important step in designing such novel membranes.

Following the Reviewer's comment, we have included this brief discussion in the main text's 'Discussion' section.

REVIEWERS' COMMENTS

Reviewer #1:

Remarks to the Author:

The authors have addressed all issues raised by this reviewer.

The publication of the manuscript is recommended.

I

Reviewer #2:

Remarks to the Author:

The authors addressed all concerns raised during the initial review process and enhanced the overall clarity of the manuscript. I recommend the acceptance of the revised manuscript for publication.

Reviewer #3:

Remarks to the Author:

The same group just published a paper titled by "Proton transport through nanoscale corrugations in two-dimensional crystals" in Nature (620, 782-786, 2023), which seems similar with this article. The novelty of this work especially the difference with (new academic contribution) the above one needs to be clarified.

RESPONSE TO REVIEWERS' COMMENTS

Reply to comment from Reviewer #1

The authors have addressed all issues raised by this reviewer.

The publication of the manuscript is recommended.

|

We are grateful with the Reviewer for recommending our manuscript for publication.

Reply to comment from Reviewer #2

The authors addressed all concerns raised during the initial review process and enhanced the overall clarity of the manuscript. I recommend the acceptance of the revised manuscript for publication.

We thank the Reviewer for recommending our manuscript for publication.

Reply to comment from Reviewer #3

The same group just published a paper titled by “Proton transport through nanoscale corrugations in two-dimensional crystals” in Nature (620, 782-786, 2023), which seems similar with this article. The novelty of this work especially the difference with (new academic contribution) the above one needs to be clarified.

The Reviewer asks us to comment on the relation between the present work and our recently published paper in Nature (Ref. 11). The results of Ref. 11 served as one of the motivations of the presented research (as discussed in the introduction), as a reference for the used technique to test the local proton conductivity of graphene oxide (SECCM) and to provide comparison with other 2D crystals (section ‘Spatial distribution of proton currents’). The present work is all about the properties of GO whereas Ref. 11 was about the role of nanoscale corrugations in graphene that are only invoked in the present paper as an explanation. Ref. 11 does not even mention graphene oxide. The conceptual overlap is minimal. Nevertheless, following the Reviewer comment we cite Ref. 11 more extensively, in all relevant places.